# IL-4 and Brentuximab Vedotin in Mycosis Fungoides: A Perspective on Potential Therapeutic Interactions and Future Research Directions

**DOI:** 10.3390/cimb47080586

**Published:** 2025-07-24

**Authors:** Mihaela Andreescu, Sorin Ioan Tudorache, Cosmin Alec Moldovan, Bogdan Andreescu

**Affiliations:** 1Department of Clinical Sciences, Hematology, Faculty of Medicine, Titu Maiorescu University of Bucharest, 040051 Bucharest, Romania; tevetmihaela@gmail.com; 2Department of Hematology, Colentina Clinical Hospital, 020125 Bucharest, Romania; 3Department of Preclinical Disciplines, Faculty of Medicine, Titu Maiorescu University, 040051 Bucharest, Romania; sorin.tudorache@prof.utm.ro; 4Department of General Surgery, Witting Clinical Hospital, 010243 Bucharest, Romania; 5Department of Medical Surgical Disciplines, Faculty of Medicine, Titu Maiorescu University of Bucharest, 031593 Bucharest, Romania; 6Department of Plastic Surgery, Colentina Clinical Hospital, 020125 Bucharest, Romania; doctorandreescu@yahoo.com

**Keywords:** mycosis fungoides, IL-4, brentuximab vedotin, CD30, tumor microenvironment, Th2 cytokines, therapeutic resistance, combination therapy, lymphoma

## Abstract

Background: Mycosis fungoides (MF), the most prevalent cutaneous T cell lymphoma, features clonal CD4⁺ T cell proliferation within a Th2-dominant microenvironment. Interleukin-4 (IL-4) promotes disease progression while Brentuximab Vedotin (BV), an anti-CD30 antibody–drug conjugate, shows efficacy but faces resistance challenges. Methods: We conducted a narrative literature review (2010–2024) synthesizing evidence on IL-4 signaling and BV’s efficacy in MF to develop a theoretical framework for combination therapy. Results: IL-4 may modulate CD30 expression and compromise BV’s effectiveness through immunosuppressive microenvironment remodeling. Theoretical mechanisms suggest that IL-4 pathway inhibition could reprogram the microenvironment toward Th1 dominance and restore BV sensitivity. However, no direct experimental evidence validates this combination, and safety concerns including potential disease acceleration require careful evaluation. Conclusions: The proposed IL-4/BV combination represents a biologically compelling but unproven hypothesis requiring systematic preclinical validation and biomarker-driven clinical trials. This framework could guide future research toward transforming treatment approaches for CD30-positive MF by targeting both malignant cells and their immunologically permissive microenvironment.

## 1. Introduction

Mycosis fungoides (MF) is the most common type of cutaneous T cell lymphoma (CTCL), accounting for approximately 54% to 65% of CTCLs. Furthermore, MF accounts for approximately 40% of cutaneous lymphomas [1]. This rare type of non-Hodgkin lymphoma is characterized by abnormal proliferation of mature, CD4-positive T cells that predominantly infiltrate the skin [2]. These malignant T cells have a great affinity for the epidermis, causing chronic and progressive skin lesions. In its early stages, MF manifests as erythematous, scaly patches or thin plaques of varying size and form, frequently developing on sun-protected areas such as the trunk and upper thighs [3]. As the disease advances, these lesions may thicken, consolidate, and finally form elevated tumors, indicating the more advanced stages of MF [4]. At the time of initial diagnosis, 70% of patients have early-stage disease (IA–IIA) [5]. MF most commonly affects individuals between 50 and 60 years of age and is more frequent in males, with a male to female ratio of approximately 2:1 [3,6]. Incidence may vary by geography and ethnicity, likely reflecting a combination of genetic, environmental, and healthcare-access factors [7,8]. Early-stage MF generally carries a favorable prognosis, with 5-year survival rates ranging from 79% to 92% [9]. Improvements in diagnostic precision have contributed to earlier detection and better outcomes. Despite these advances, MF remains challenging to diagnose and manage because of its varied clinical presentation and slow, often unpredictable progression [1]. The pathogenesis of MF is a multifaceted interaction of genetic abnormalities, microenvironmental variables, and immunological dysregulation. T cells, particularly CD4-positive cells, constitute nearly two-thirds of all cutaneous lymphomas, and this is the most common immunophenotype seen in MF [10]. The disease is marked by a clonal expansion of CD4^+^ T cells that often lack typical surface markers such as CD2, CD5, or CD7 [11]. Keratinocytes play a key role in attracting these malignant T cells to the skin. Once in the dermis, the neoplastic T cells accumulate around Langerhans cells, which lead to the formation of Pautrier microabscesses. As the disease progresses, some malignant cells migrate to regional lymph nodes and subsequently enter the bloodstream and circulate alongside other cutaneous lymphocyte antigen (CLA)-positive T cells [12].

Despite multiple available therapies, no curative treatment exists, and MF remains largely incurable. Treatment is often complicated by the chronic nature of the disease, frequent relapses, and the significant impact on patients’ quality of life. Furthermore, MF can resemble other dermatologic disorders which may delay both diagnosis and appropriate treatment [13]. Therapies are typically stage-directed: early disease is managed with skin-directed approaches such as topical corticosteroids and phototherapy, while advanced or refractory MF requires systemic treatments such as retinoids, interferon-α, chemotherapy, histone deacetylase inhibitors, and monoclonal antibodies [14]. However, responses are often incomplete and transient. Recent trials have introduced targeted agents such as the anti-CD30 antibody–drug conjugate (ADC) Brentuximab Vedotin (BV) for CD30-expressing MF [15]. However, disease relapses and symptom burden remain major challenges. The complex MF microenvironment, including its Th2 cytokine profile, contributes to treatment resistance. IL-4 and related Type 2 cytokines are thought to drive tumor growth and immune evasion in MF [16]. Given this, there is growing interest in understanding how IL-4 signaling intersects with new therapies like BV, especially in CD30 MF.

As clinicians, we routinely encounter patients who illustrate what has already been established: MF is both frequent and difficult to treat effectively in the long term. In advanced or treatment-refractory cases, median survival can drop to 2–4 years, with quality of life significantly impaired by skin failure, infections, or systemic symptoms. The need to expand available therapeutic options is not theoretical—it is urgent and ongoing. This review was motivated by the observed intersection between IL-4-driven immune dysregulation and the partial success of CD30-targeted therapy in MF. Despite the potential for interaction, these two axes have never been explored in tandem. Here, we aimed to provide a mechanistic rationale to support future investigations of this untested therapeutic combination.

## 2. Methods

We conducted a narrative review of the PubMed, Embase, and Cochrane databases (between January 2010 and December 2024) using the following search terms and Boolean operators: (“mycosis fungoides” OR “cutaneous T cell lymphoma”) AND (“interleukin-4” OR “IL-4”) AND (“Brentuximab Vedotin” OR “CD30”). Additional searches included combinations with “tumor microenvironment,” “Th2”, and “dupilumab” and the language was restricted to English. Articles were screened for relevance to MF pathogenesis, IL-4 signaling, and CD30-targeted therapy. Given the absence of direct evidence for IL-4/BV combination therapy, we synthesized findings from related studies to develop the theoretical framework presented.

## 3. Results

### 3.1. IL-4 in MF Pathogenesis

#### 3.1.1. IL-4 in General Cancer Biology

Interleukin-4 (IL-4) is a cytokine primarily produced by immune cells such as eosinophils, basophils, and T-helper 2 (Th2) cells. IL-4 exerts broad effects on tumor biology by modulating the tumor microenvironment (TME) and promoting immune evasion. The principal function of IL-4 is to drive the differentiation of naive CD4^+^ T cells into Th2 cells through IL-4-mediated signaling, ultimately altering the adaptive immune response [17,18]. This process not only shifts immune polarization but also suppresses the Th1 and Th17 responses that are critical for effective anti-tumor activity.

IL-4 contributes to tumor growth via several mechanisms, including reducing interferon-gamma (IFN-γ) production and upregulating anti-apoptotic genes in tumor cells, supporting their survival and growth [19,20]. It also promotes macrophage polarization toward the M2 phenotype, which enhances tumor proliferation and suppresses anti-tumor immunity, and facilitates the expansion of myeloid-derived suppressor cells (MDSCs), further amplifying immunosuppression within the TME [21].

By inducing a Th2-skewed immune response and increasing regulatory T cell activity, IL-4 suppresses the cytotoxic function of CD8^+^ T cells—an effect that undermines the efficacy of immunological checkpoint therapies. This resistance mechanism provides a strong rationale for combining IL-4 pathway inhibitors with immune checkpoint blockades, with the goal of restoring anti-tumor immunity and improving treatment outcomes, particularly in CD30-positive lymphomas such as mycosis fungoides. The presence and functional state of immune cells within the tumor microenvironment thus remain critical determinants of a therapeutic response.

Clinical data show that chronic inflammation at tumor sites is often accompanied by elevated IL-4 levels, and IL-4-producing CD8^+^ T cells are less effective at tumor clearance compared with their non-IL-4-producing counterparts. Moreover, IL-4 secreted by cancer cells can mediate resistance to immune checkpoint blockade (ICB) therapies, reducing the efficacy of treatments such as PD-1/PD-L1 inhibitors [21,22,23]. This highlights the potential benefit of combining IL-4 pathway inhibitors with ICB or other immunotherapies such as Brentuximab Vedotin to overcome tumor immune evasion [24].

Notably, IL-4’s role in cancer is context-dependent. While generally immunosuppressive, under certain circumstances, IL-4 can paradoxically enhance anti-tumor immunity. The outcome depends on factors such as the type of IL-4 receptor engaged (Type I: IL-4Rα/γc vs. Type II: IL-4Rα/IL-13Rα1) and the presence of regulatory molecules such as SOCS1 and SOCS3, which can modulate downstream STAT6 signaling and explain heterogeneous responses among patients [25].

Recent work with engineered IL-4 variants suggests that selective modulation of IL-4 signaling could revitalize exhausted CD8^+^ T cells, providing new opportunities to enhance anti-tumor immunity while minimizing the immunosuppressive effects of the cytokine [26].

#### 3.1.2. IL-4 in MF-Specific Pathogenesis

Mycosis fungoides (MF), the most common cutaneous T cell lymphoma, is characterized by a Th2-skewed immune microenvironment [27]. As MF progresses, there is a decline in anti-tumor Th1 responses (e.g., IFN-γ, IL-12) and a predominance of Th2 cytokines such as IL-4, IL-5, and IL-13 [16,28]. Elevated IL-4 production by malignant and reactive immune cells is observed even in early MF [29,30,31].

IL-4 promotes the development of naive CD4^+^ T cells into Th2 cells, yielding a cytokine profile dominated by IL-4, IL-5, and IL-13 [32]. This transition undermines cytotoxic anti-tumor immunity and fosters a tumor-permissive microenvironment. By directly suppressing Th1 cytokines and promoting MF tumor cell survival and proliferation (often in conjunction with IL-13) [33], IL-4 enables immune evasion, chronic inflammation, and fibrosis in the skin [28].

IL-4-driven macrophage polarization toward the immunosuppressive M2 phenotype is prominent in MF. M2-like tumor-associated macrophages (TAMs), which are abundant in MF lesions, secrete high levels of IL-4, IL-13, and TGF-β. Kim et al. showed that M2 TAMs (CD163+) dominate the MF microenvironment, constituting 40% of the infiltrate and contributing to immune evasion; these macrophages may also co-express CD30 [34,35]. Clinically, elevated IL-4, IL-13, and IL-31 are implicated in the pruritus seen in MF, with IL-31 levels correlating with both disease progression and symptom severity [28]. Accordingly, blockade of the IL-4/IL-13 axis (e.g., dupilumab) has been investigated for symptom control and potential disease modification.

Additional evidence links IL-4 to MF-specific stromal remodeling. Elevated IL-4 levels stimulate dermal fibroblasts to produce periostin, which drives the production of thymic stromal lymphopoietin, activating STAT5 and further amplifying IL-4 synthesis [16,36]. Recent studies underscore the central role of Th2 cytokines, particularly IL-4 and IL-13, in immune evasion within MF. For example, Guenova et al [37] found that malignant lymphocytes frequently produce high levels of Th2 cytokines but lack IFNγ expression, a hallmark of impaired immune surveillance and unchecked tumor growth. 

Strategies to counteract this skewed cytokine milieu include targeting internal signaling with agents such as JAK inhibitors (e.g., ritlecitinib [38]) and blocking the IL-4 axis with IL-4Rα or IL-13 antibodies, which are currently under investigation in CTCL [28]. However, preclinical and clinical studies with Th2-targeted therapies, including JAK inhibition and PD-1 blockade [36], have yielded mixed results, with some interventions paradoxically accelerating tumor proliferation. These findings highlight both the promise and the complexity of restoring immune defense mechanisms in MF by modulating Th2 polarization. A nuanced understanding of IL-4’s role remains essential for designing combination treatments that effectively target both malignant cells and the Th2-skewed microenvironment. 

It is important to note that while targeting the IL-4 axis holds therapeutic promise, there are reports of MF’s acceleration following dupilumab initiation [39,40,41]. Most such cases involve unrecognized pre-existing MF rather than true induction, and the incidence appears very low. Careful patient selection and monitoring are required, especially in patients presenting with atypical eczema or recalcitrant dermatitis.

#### 3.1.3. Molecular Mechanisms: The IL-4/STAT6/CD30 Axis

When IL-4 binds to the IL-4 receptor (IL-4R) on target cells, it activates the downstream Janus kinase (JAK) and signal transducer and activator of transcription 6 (STAT6) pathways [42]. The IL-4/STAT6 axis plays a pivotal role in tumor development and the establishment of an immunosuppressive milieu [42,43,44]. By upregulating anti-apoptotic genes and inhibiting Th1/Th17 differentiation, STAT6 activation supports the survival and proliferation of malignant T cells [16].

Recent evidence in follicular lymphoma further demonstrates that activating mutations in STAT6 sensitize tumor cells to IL-4, driving downstream mTOR pathway activation and compensating for CREBBP deficiency. This highlights the oncogenic importance of IL-4/STAT6 pathway dysregulation across multiple lymphoma subtypes, not only MF [45].

Upon activation, STAT6 induces the expression of several anti-apoptotic genes, notably Bcl-2 and Bcl-xL. These proteins inhibit programmed cell death and contribute to the persistence and chemoresistance of malignant T cells. Upregulation of Bcl-2 and Bcl-xL via IL-4/STAT6 signaling is therefore a key mechanism by which tumor cells evade apoptosis and sustain their growth under therapeutic pressure.

A further layer of complexity is added by the positive feedback between NLRP3 and the IL-4 promoter, particularly in malignant CD4^+^ T cells of cutaneous T cell lymphoma, which perpetuates high IL-4 production and reinforces the Th2-dominant, immunosuppressive environment [46].

Central to the discussion of combinatorial therapy in MF is the relationship between IL-4 signaling and CD30 expression. IL-4 can modulate CD30 expression through the STAT6 pathway in a context-dependent manner. In some settings, IL-4 upregulates CD30 expression on activated CD4^+^ T cells, thereby increasing the density of BV targets on the tumor surface [47]. However, other studies have shown that sustained or high-level IL-4/STAT6 signaling can downregulate CD30, potentially by inducing inhibitory transcription factors or altering chromatin accessibility at the CD30 promoter [48]. 

The effect of IL-4 on CD30 is context-dependent. Factors such as cell type, exposure timing, and local signaling cues determine whether CD30 is up- or downregulated. This variability directly affects how well CD30-targeted therapies like Brentuximab Vedotin perform, influencing both the initial drug response and the emergence of resistance.

Temporal aspects are also critical: chronic IL-4 exposure induces SOCS proteins that dampen STAT6 signaling, while acute exposure may transiently increase IL-4R expression, impacting the timing and effectiveness of IL-4 blockade relative to BV administration.

A comprehensive understanding of the IL-4/STAT6/CD30 axis is therefore essential (Figure 1) for rational combination therapy design in MF, as it may inform both biomarker selection and the timing of interventions (Figure 2).

### 3.2. Brentuximab Vedotin in MF

Brentuximab Vedotin is an antibody–drug conjugate (ADC) that targets cells that express CD30, which is variably expressed in various lymphomas, including subsets of MF (Figure 3). The ADC is made up of a chimeric anti-CD30 monoclonal antibody linked to monomethyl auristatin E (MMAE), a potent antimitotic chemical [49]. After binding to CD30 on malignant cell surfaces, the Brentuximab Vedotin–CD30 complex is internalized via clathrin-mediated endocytosis. The combination is then transferred to the lysosomes, where proteolytic breakdown releases MMAE into the cytoplasm. Free MMAE interacts with tubulin, disrupting the microtubule network, causing cell cycle arrest in the G2/M phase and apoptosis. This targeted dosing increases the cytotoxic effect on malignant cells while lowering systemic toxicity [48].

In the pivotal ALCANZA trial, BV demonstrated significantly superior efficacy to standard treatments in CD30-positive MF, with a 56% ORR4 and improved progression-free survival [50,51]. Peripheral neuropathy was the most common adverse event [52]. These data firmly position BV as a standard-of-care option in CD30-positive MF and CTCLs, though toxicity—particularly neuropathy—remains a limiting factor.

Subsequent studies, including a Phase II trial by Kim et al., confirmed high response rates across CD30 expression levels, with better responses in CD30-high lesions [35]. Recent real-world analyses reinforce BV’s efficacy. Real-world studies reinforce BV’s superiority over older systemic therapies, with sustained responses and improved time to next treatment [53]. Time to next treatment and progression-free survival were significantly longer with BV than with other standard therapies (methotrexate, mogamulizumab, bendamustine, etc.). Overall survival at 1–2 years was also improved. This “real-world” experience aligns with the clinical trials, indicating that BV substantially outperforms older systemic options in CD30^+^ MF patients [54].

### 3.3. Theoretical Framework for Combination Therapy

The interaction between IL-4 signaling and Brentuximab Vedotin’s efficacy in mycosis fungoides (MF) reflects a convergence of tumor immunobiology and targeted therapy. Although IL-4 and CD30 are components of distinct immune pathways, they may intersect in ways that affect the treatment response. 

The Th2-dominant tumor microenvironment (TME) in MF features elevated IL-4 and IL-13, which promote malignant T cell survival and suppress cytotoxic responses [55,56]. IL-4 also drives M2 macrophage polarization and regulatory T cell recruitment, further enabling immune evasion [57,58]. In contrast, Brentuximab Vedotin (BV) has shown durable efficacy in CD30-positive MF, underscoring its therapeutic value despite the immunosuppressive milieu. BV’s efficacy is impacted by the heterogeneity of CD30 expression in MF, which varies significantly among patients and even within individual tumors, emphasizing the need for biomarker-driven patient selection [59].

Data regarding IL-4’s effect on CD30 expression remain contradictory: some studies demonstrate upregulation [60], while others report downregulation mediated by STAT6 signaling [61,62]. This variability highlights the importance of the cellular context and may help explain the heterogeneous clinical responses observed in MF. For example, IL-4-induced upregulation of CD30 on activated CD4^+^ T cells [63] suggests that high IL-4 concentrations within the MF microenvironment could theoretically enhance the BV target’s density and binding efficiency [55]. However, these potential benefits may be offset by IL-4’s immunosuppressive effects. Importantly, there is currently no direct evidence that modulation of IL-4 improves BV’s outcomes in MF, and the existing mechanistic rationale is largely hypothetical, based on indirect observations from other disease settings [61,62]. Further studies are needed to clarify how IL-4 influences CD30 dynamics and the clinical efficacy of BV in MF.

Conversely, IL-4 has also been reported to downregulate CD30 via STAT6 signaling, potentially reducing BV binding and efficacy [62]. These conflicting observations suggest that IL-4 may exert both positive and negative effects on CD30 expression, depending on the cellular context. Further study is needed to clarify IL-4’s net influence on the BV response in CD30+ lymphomas, including MF [62]. This contradiction reflects the complex and context-dependent regulation of CD30. It may be influenced by the differentiation state of the cells, concurrent signaling pathways, the duration and intensity of IL-4 exposure, and the surrounding cytokine activity. While direct data on this interaction in human MF remain limited, these findings open avenues for further investigation. 

Beyond CD30 regulation, Th2 cytokines may also affect BV’s function by altering target expression, antibody internalization, or intracellular trafficking—though direct evidence in MF is lacking. IL-4-driven Th2 polarization reshapes the TME, reducing cytotoxic T cell infiltration, increasing regulatory T cells and tumor-associated macrophages, and impairing immune surveillance. This may hinder BV’s cytotoxic activity via impaired ADCC and ADCP, potentially compromising drug delivery and tumor recognition. IL-4 has been linked to the emergence of resistance mechanisms to many medications [63]. For BV, IL-4-induced downregulation of CD30 expression may cause partial resistance due to a reduction in the number of target receptors available for drug binding [62]. This resistance can be overcome by combining Brentuximab Vedotin with other drugs such as masitinib or PKC412. Furthermore, IL-4 facilitates immune evasion by upregulating anti-apoptotic proteins such as Bcl-2 and Bcl-xL, promoting tumor cells’ survival and resistance to apoptosis. This can protect tumor cells from apoptotic stimuli induced by chemotherapeutic agents. IL-4-induced survival signals via JAK-STAT6 may counteract MMAE-induced apoptosis, even when the drug successfully enters malignant cells.

Intriguingly, the clinical success of IL-4 pathway inhibitors (such as dupilumab, which blocks IL-4Rα, or various JAK inhibitors) in managing Th2-driven inflammatory diseases like atopic dermatitis and asthma provides a compelling precedent. While distinct from lymphomas, their ability to modulate the Th2 axis stimulates considerable interest in their role within the Th2-skewed TME of MF. 

The pursuit of this promising combination is tempered by concerning signals that mandate a cautious approach. As discussed in Section 3.2, IL-4 signaling may have dual roles in MF, and its modulation requires careful patient selection and monitoring.

Based on these insights and preliminary observations, theoretical proposals suggest a synergistic potential for dual targeting. Inhibiting the IL-4 pathway could enhance BV’s efficacy by reversing the immunosuppressive effects of IL-4, such as M2 macrophage polarization and T cell suppression. This restoration of immune surveillance could create a more favorable anti-tumor environment for BV to exert its cytotoxic effects, leading to more profound and durable responses. The shift from Th2 to Th1 dominance could restore cytotoxic immunity, reduce IL-4-mediated survival signals in malignant cells, stabilize or enhance CD30 expression, and improve overall immune surveillance and ADCC mechanisms.

A combined therapeutic strategy might help overcome the known mechanisms of resistance to BV. Primary resistance may stem from low baseline CD30 expression or IL-4-driven downregulation that limits BV binding. Over time, chronic IL-4 exposure could select for CD30-low clones or trigger adaptive mechanisms that reduce BV sensitivity. Additionally, the microenvironmental protection provided by IL-4-driven M2 macrophages and regulatory T cells may create physical and immunological barriers that protect malignant cells from BV-induced cytotoxicity.

Since IL-4 contributes to PD-L1 upregulation and T cell exhaustion, triple therapy combining BV, IL-4 blockade, and PD-1/PD-L1 inhibition may offer synergistic benefits. JAK inhibitors can block IL-4/STAT6 signaling while affecting multiple cytokine pathways, offering an alternative to specific IL-4 blockade. Some evidence suggests histone deacetylase inhibitors may modulate both CD30 expression and cytokine production, potentially complementing the IL-4/BV interaction.

Realizing this potential depends on biomarker-guided patient selection, which we will further discuss in the next section. 

While a strong mechanistic foundation supports IL-4/BV co-targeting, key uncertainties persist. Optimal sequencing of IL-4 blockade relative to BV administration is yet unknown—should IL-4 inhibition precede, accompany, or follow BV treatment? Beyond CD30 expression, what clinical or molecular features predict a benefit from dual targeting? The combined toxicity profile requires careful evaluation. Brentuximab Vedotin is notoriously associated with a significant incidence of peripheral neuropathy, which can affect up to 67% of patients in some studies, which impacts both dose-limiting and patient quality of life. The potential for cumulative or synergistic neurotoxicity when combining BV with other agents remains an unknown that must be meticulously evaluated. Whether IL-4 blockade can prevent or delay the development of BV resistance remains speculative.

Perhaps the most significant impediment to immediate clinical translation is the absolute lack of direct clinical evidence. There are no published clinical studies evaluating the safety or efficacy of combining IL-4 pathway inhibitors with Brentuximab Vedotin in MF patients. All potential benefits discussed remain purely theoretical or derived from indirect experimental evidence. The strong mechanistic rationale and preliminary observations warrant systematic investigation through carefully designed preclinical studies and biomarker-driven clinical trials. The journey from evidence-informed hypothesis to validated clinical practice will require attention to both efficacy and safety, but the potential to improve outcomes for patients with CD30-positive MF warrants further investigation.

The IL-4/BV combination should be considered alongside other emerging strategies:BV + checkpoint inhibitors: Phase II trials combining BV with pembrolizumab show ORR of 71% in relapsed CTCL;BV + lenalidomide demonstrates synergistic activity through immunomodulation;Mogamulizumab combinations: anti-CCR4 therapy depletes Tregs, potentially complementing IL-4 blockade;CAR-T approaches: CD30-targeted CAR-T cells are in development, raising questions about sequencing with BV.

### 3.4. Evidence Gaps and Research Priorities

#### 3.4.1. Knowledge Gaps 

Despite the biologically supported case for combining IL-4 inhibition with Brentuximab Vedotin in mycosis fungoides, its clinical translation is currently hindered by several critical knowledge gaps. These gaps underscore the urgent need for focused research to bridge the divide between theoretical promise and clinical applicability.

The absence of any preclinical or clinical data directly testing IL-4 inhibition with BV represents the most fundamental gap in our understanding. Specific experimental models urgently needed include: MF-derived cell lines (e.g., HH, MJ, HuT-78) treated with IL-4 ± anti-IL-4Rα antibodies to assess CD30’s expression dynamics;Patient-derived xenograft (PDX) models testing combinations’ efficacy;Ex vivo cytotoxicity assays using patient samples to measure ADCC/ADCP under IL-4 modulation.

For a combination strategy to be clinically relevant, any IL-4-mediated modulation of CD30 needs to be rapid and sustained. 

It remains unclear how IL-4 signaling—or its inhibition—affects CD30 expression in human MF cells. This includes both direct effects and broader microenvironmental influences. These mechanisms likely extend beyond simple receptor-level regulation. We need to determine the exact molecular pathways involved in any such modulation and, importantly, the kinetics of these effects to understand whether IL-4’s effects on CD30 are direct transcriptional modifications or indirect consequences of broader cellular state changes. 

Beyond mere CD30 expression, the precise criteria for selecting the MF patients who would most optimally benefit from this combination therapy are undefined. There is an urgent need to identify robust predictive biomarkers—whether they are specific cytokine profiles, patterns of immune cell infiltrates, or underlying genetic mutations—that can reliably predict response or resistance to the proposed combination. Current CD30 expression thresholds for BV treatment vary widely in clinical practice, and adding IL-4 pathway status as another variable exponentially increases the complexity of patient selection. Molecular biomarker candidates requiring validation include:Serum markers: IL-4 (>50 pg/mL), IL-13 (>100 pg/mL), and soluble IL-4Rα (>500 ng/mL) measured by validated ELISA platforms;Tissue markers: STAT6 phosphorylation (pSTAT6) by immunohistochemistry (H-score >100), GATA3 expression (>30% nuclear positivity), and CD163+ M2 macrophage density (>40% of infiltrate);Genetic signatures: the 15-gene IL-4 response signature by NanoString or RT-PCR, with scores >75th percentile indicating high pathway activity;Functional assays: ex vivo IL-4-induced proliferation index >2.0 in patients’ T cells;CD30 thresholds: minimum 10% expression by IHC for BV eligibility, with exploration of whether IL-4 inhibition could rescue patients with 5–10% expression.

The optimal treatment sequencing and dosing regimens for these two agents are entirely unknown. Crucial questions revolve around whether IL-4 pathway inhibition should precede, follow, or be co-administered concurrently with BV, and what doses and schedules will maximize efficacy while concurrently minimizing toxicity. Should IL-4 blockade be used as a "priming" strategy to optimize CD30 expression before BV administration? Or should it be maintained throughout BV treatment to prevent adaptive resistance? These questions have profound implications for clinical trial design and eventual treatment protocols.

It is still unclear how IL-4 pathway modulation—when combined with BV—affects anti-tumor immunity and long-term surveillance. This question is central to predicting long-term disease control and patient outcomes.

Will combination therapy lead to more durable remissions by preventing the re-establishment of the immunosuppressive TME? Or might it inadvertently create new mechanisms of immune escape?

A critical gap exists in our understanding of how tumors might develop resistance to combined IL-4/BV therapy. Potential resistance mechanisms include compensatory increases in IL-13 or IL-5 after IL-4 blockade, selection for CD30-negative clones, activation of alternative survival pathways, altered MMAE metabolism or drug efflux, and TME remodeling that limits antibody access.

The pharmacokinetic and pharmacodynamic interactions between IL-4 pathway inhibitors and BV remain unexplored. Key questions include whether IL-4 blockade affects the biodistribution or clearance of BV, whether there are drug–drug interactions at the level of hepatic metabolism, how the different half-lives of these agents impact optimal dosing schedules, and what the impact of combination therapy is on normal tissue that expresses both CD30 and IL-4 receptors.

MF encompasses a spectrum of clinical presentations and molecular subtypes. Critical gaps in our understanding include whether folliculotropic, syringotropic, or other MF variants respond differently to combination therapy; the role of disease stage (patch, plaque, tumor) in determining treatment response; how transformed MF might behave differently, given its altered biology; and the impact of prior therapies on the IL-4/CD30 axis and subsequent response to combination treatment.

Defining the most appropriate primary and secondary endpoints for clinical trials evaluating this combination is complex, especially given the chronic and often indolent nature of MF, where traditional endpoints may not fully capture the long-term benefit. Should trials focus on response rates, progression-free survival, or quality of life measures? How can we account for the potential delayed benefits of immunomodulation?

The impact of combination therapy on long-term outcomes remains entirely speculative. Key unknowns include the durability of responses achieved with combination therapy, the risk of secondary malignancies or long-term immune dysfunction, the quality of life implications of prolonged dual pathway inhibition, and the potential for treatment-free remission or functional cure in some patients.

As the therapeutic landscape for MF continues to evolve, understanding how IL-4/BV combination therapy might integrate with other emerging treatments becomes crucial. Gaps include optimal combinations with checkpoint inhibitors, CAR-T cells, or other immunotherapies; sequencing with other targeted agents like mogamulizumab or HDAC inhibitors; the role of combination therapy in stem cell transplant eligibility or outcomes; and integration with novel skin-directed therapies or radiation approaches.

These critical evidence gaps collectively highlight the substantial work required to translate the promising biological rationale of IL-4/BV combination therapy into clinical practice. Addressing these gaps systematically through well-designed preclinical studies, biomarker development programs, and carefully conducted clinical trials will be essential for determining whether this approach can fulfill its theoretical promise of improving outcomes for patients with CD30-positive mycosis fungoides.

To address the current gap in the literature we performed a simple search in PubMed, Embase, and Web of Science for clinical trials, cohort or case–control studies, case series, mechanistic laboratory studies, and systematic or narrative reviews published from January 2008 to June 2024, with search strings including combinations of the following keywords and MeSH terms: “mycosis fungoides” OR “cutaneous T cell lymphoma” OR “CTCL” and “interleukin-4” OR “IL-4”; “Brentuximab Vedotin” OR “BV” OR “anti-CD30 antibody”, restricted to English-language articles including human studies or relevant preclinical models of MF or CTCL, studies evaluating IL-4 signaling, Brentuximab Vedotin, or both. Filtered results revealed a small number of relevant articles, but none addressing a combined approach of BV and IL-4 inhibition. 

A summary of the knowledge gaps and the research questions they lead to is summarized in Table 1.

#### 3.4.2. Proposed Research Framework: A Phased Approach to Clinical Translation

Addressing the critical evidence gaps previously identified necessitates a structured, multi-phased research framework, meticulously designed to guide the translation of this promising hypothesis from a fundamental mechanistic understanding through to rigorous clinical validation and, ultimately, definitive evaluation. This systematic approach is crucial to ensure both efficacy and patient safety.

The initial phase must rigorously establish a mechanistic foundation and conduct preclinical validation. This will involve comprehensive in vitro studies utilizing established MF cell lines and, critically, primary patients’ MF cells and skin explants. These experiments should be designed to precisely elucidate how IL-4 signaling and its inhibition affect CD30 expression dynamics, BV internalization, MMAE delivery, and the subsequent apoptotic pathways within MF cells. Concurrently, it is vital to investigate the impact on MF cells’ proliferation and survival, and their complex interactions with various immune cell subsets (e.g., macrophages, T cells) within a relevant microenvironmental context. Furthermore, the utilization of sophisticated in vivo animal models that accurately mimic the human disease and its TME is essential. These models will assess the combination’s efficacy, safety, and pharmacologic profile—especially the risk of neurotoxicity. Importantly, preclinical work must also evaluate the possibility of paradoxical effects, including disease acceleration. This concern has emerged with IL-4 inhibitors in some clinical contexts.

Crucially, preclinical studies should also explore the potential for paradoxical effects or disease acceleration, as has been an observed concern with IL-4 pathway inhibitors in some contexts. Parallel to these mechanistic studies, a strong emphasis should be placed on robust biomarker discovery and early validation. This includes analyzing gene expression profiles (e.g., via RNA sequencing), cytokine arrays, and immune cell profiling (e.g., flow cytometry, immunohistochemistry) from patient biopsies and peripheral blood. The ultimate aim is to identify measurable indicators of IL-4 pathway activation and CD30 status that can reliably predict the response to both single-agent therapies and, subsequently, to the combination, guiding future patient selection.

Following promising preclinical signals, the research should progress to early clinical validation and comprehensive biomarker collection. This phase would involve the initiation of carefully designed dose-escalation (Phase I) and proof-of-concept (Phase II) clinical trials in patients with CD30-positive, relapsed/refractory MF. The primary focus of these trials would be to rigorously assess the safety and tolerability of the combination, meticulously monitoring adverse events. Patient stratification must be rigorously guided by the current understanding of CD30 expression and, ideally, by the early biomarkers of IL-4 pathway activity identified in the preclinical phase. Comprehensive biomarker collection from pre- and on-treatment biopsies and peripheral blood samples is essential in this phase. This includes assessing precise changes in CD30 expression, the dynamics of IL-4 pathway activation, the composition of the TME, and alterations in immune cell subsets. These data will be crucial for correlating molecular changes with clinical responses and for identifying robust predictive factors. Furthermore, these early-phase trials should also aim to determine optimal dosing regimens and potential sequencing strategies (e.g., concurrent versus sequential administration) that maximize therapeutic efficacy while concurrently minimizing toxicity.

Finally, should the combination therapy demonstrate compelling safety and efficacy signals in the early clinical phases, the definitive evaluation and real-world implementation phase would commence. This stage necessitates larger, well-designed randomized controlled trials (RCTs). These RCTs would rigorously compare the combination therapy against single-agent BV (or other established standard-of-care options) in meticulously defined patient populations. Key endpoints for these trials would extend beyond traditional response rates to include objective response rates, the duration of response, progression-free survival, overall survival, and patient-reported outcomes (PROs) related to quality of life. Furthermore, comprehensive long-term follow-up studies will be essential to accurately assess the durability of response and to identify any delayed or cumulative toxicities that may emerge over extended treatment periods. 

This study presents a theoretical framework for combining IL-4 pathway inhibition with Brentuximab Vedotin in the treatment of CD30-positive mycosis fungoides, but several limitations must be acknowledged. First, the analysis is based on a narrative literature review, without direct experimental data or clinical evidence supporting the proposed combination. The conclusions drawn are therefore speculative and rely heavily on indirect evidence from related disease models. Second, the heterogeneity of CD30 expression in MF and the context-dependent effects of IL-4 signaling introduce biological variability that complicates therapeutic predictions. Additionally, no validated biomarkers currently exist to identify the patients most likely to benefit from IL-4/BV co-targeting. Finally, potential adverse effects, such as cumulative neurotoxicity or paradoxical disease progression observed in some IL-4 pathway inhibitor cases, remain theoretical and unquantified. These limitations highlight the need for rigorous preclinical validation and biomarker-guided clinical trials before this combination strategy can be translated into clinical practice.

## 4. Conclusions

The convergence of IL-4 signaling and Brentuximab Vedotin (BV) therapy in CD30-positive mycosis fungoides presents a compelling but as yet unvalidated therapeutic hypothesis. The biological rationale for synergy between IL-4 pathway inhibition and BV is supported by multiple mechanistic pathways: IL-4’s immunosuppressive remodeling of the tumor microenvironment, its modulation of CD30 expression, and its upregulation of anti-apoptotic signals all potentially intersect with BV’s mode of action. Inhibition of IL-4—via IL-4Rα blockade or JAK/STAT6 pathway modulation—could reprogram the microenvironment toward a Th1 phenotype, restore cytotoxic immune function, and possibly augment BV’s efficacy. Co-inhibition of immune checkpoints (e.g., PD-1/PD-L1) may further amplify this effect. However, no direct preclinical or clinical evidence currently supports the safety or efficacy of combining IL-4 pathway inhibitors with BV in MF. The interaction between IL-4 and CD30 appears highly context-dependent, and conflicting data on CD30 regulation by IL-4 underscore the complexity of this potential synergy. Safety concerns, including the risk of accelerated disease progression seen with IL-4R blockade in some MF patients and the well-documented neuropathy associated with BV, raise the stakes for any combination approach.

Despite these complexities, the hypothesis remains biologically plausible and clinically relevant. Future research should prioritize preclinical models to test the safety and efficacy of combining IL-4 pathway inhibition with BV, with a particular focus on tumor microenvironment remodeling, CD30 expression kinetics, and cytotoxic immune function. Biomarker development—including STAT6 activation, GATA3 expression, and CD30 dynamics—may further guide patient selection and treatment sequencing in early-phase trials.

## Figures and Tables

**Figure 1 cimb-47-00586-f001:**
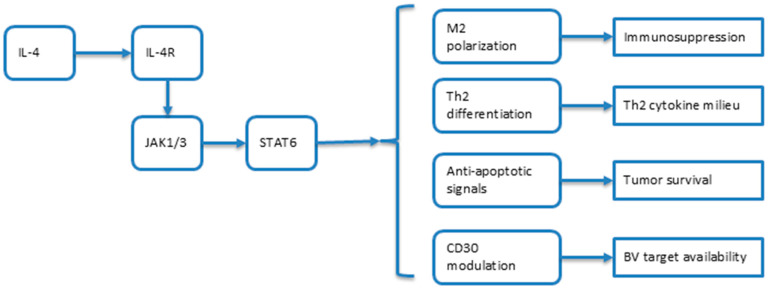
Schematic representation of the IL-4/STAT6 signaling axis in the tumor microenvironment and IL-4/13 inhibitors.

**Figure 2 cimb-47-00586-f002:**
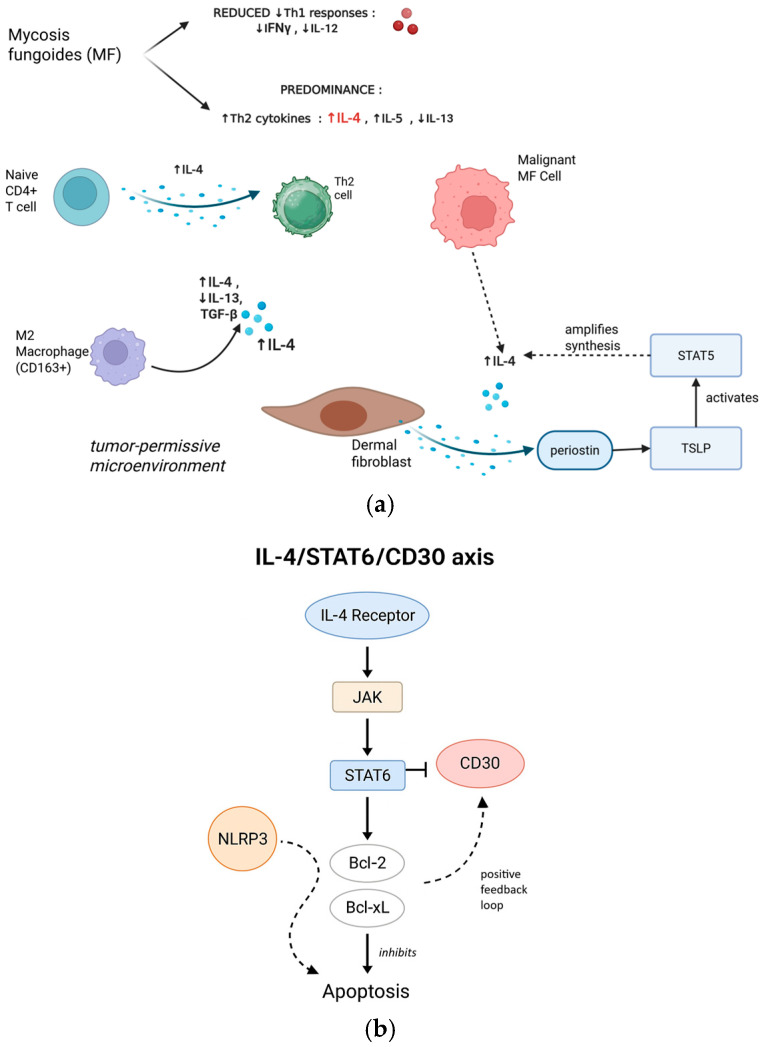
Mycosis fungoides and IL-4/13 inhibitors. (**a**) IL-4-driven Th2 polarization and microenvironment remodeling in mycosis fungoides. Increased IL-4 production from Th2 cells, M2 macrophages, and malignant MF cells promotes a tumor-permissive environment via dermal fibroblast activation, periostin synthesis, and TSLP–STAT5 signaling. (**b**) IL-4/STAT6/CD30 axis: IL-4 signaling upregulates Bcl-2 and Bcl-xL to inhibit apoptosis, with CD30 and NLRP3 reinforcing a pro-survival feedback loop.

**Figure 3 cimb-47-00586-f003:**
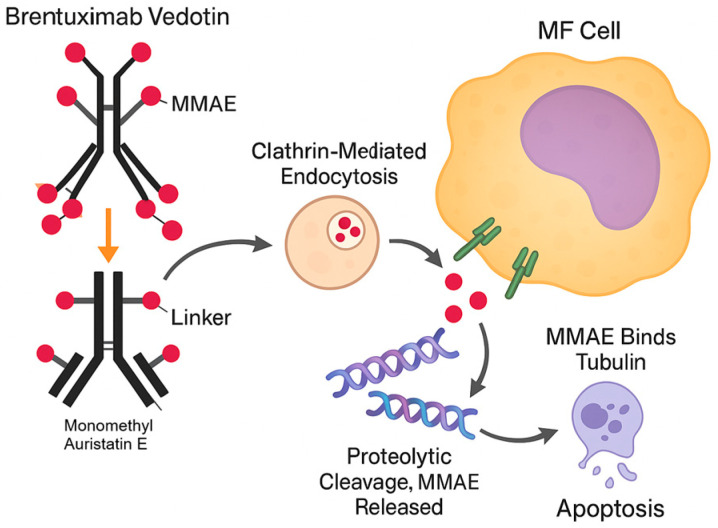
Mechanism of action of the Brentuximab Vedotin antibody–drug conjugate (ADC).

**Table 1 cimb-47-00586-t001:** Summarization of the knowledge gaps and prospective questions the current research leads to.

Knowledge Gap	Research Question	Confidence Level	Clinical Impact
Unclear whether IL-4 upregulates or downregulates CD30 in the MF context.	How does IL-4 signaling affect CD30 expression in malignant T cells from MF patients under different conditions?	Moderate	Directly affects patient selection and the likelihood of BV efficacy in combination strategies.
Impact of Th2 cytokines on ADC efficacy is poorly understood.	What is the effect of Th2 cytokines (IL-4, IL-13) on the internalization and cytotoxic activity of Brentuximab Vedotin in MF?	Moderate	May influence response rates and the durability of BV therapy; could necessitate adjunct immunomodulation.
Limited data on IL-4/IL-13 inhibitors in MF.	Do IL-4 pathway inhibitors demonstrate clinical activity or modify the disease course in MF patients?	Moderate	Limits the adoption of IL-4/IL-13 inhibitors in routine MF management or in combination regimens.
No evidence for synergy between IL-4 blockade and BV.	Does IL-4 pathway inhibition enhance the efficacy of BV in MF by modulating immune activity or altering CD30 expression?	Low	Prevents rational design of combination trials; synergy would justify new therapeutic protocols.
Resistance mechanisms in MF not fully addressed by current therapies.	Can combination therapy targeting IL-4 and CD30 overcome intrinsic or acquired resistance in MF?	Low	Novel combinations may offer solutions for refractory/relapsed MF patients.
Biomarker frameworks for combination strategies are conceptual.	What biomarkers can identify the MF patients likely to benefit from IL-4 and BV co-targeting?	Low	Limits personalized therapy and efficient trial enrollment; impedes biomarker-driven clinical practice.
Risk of MF progression with IL-4 inhibitors.	What is the mechanism behind disease acceleration in MF patients receiving dupilumab?	Requires caution	Uncertainty deters clinicians from using IL-4 blockade; may expose patients to unforeseen adverse events.
The neurotoxicity profile of BV in combinations is unknown.	What is the cumulative neurotoxicity risk of combining BV with IL-4 pathway inhibitors in MF patients?	Requires caution	Potential for increased or unexpected toxicity could limit the clinical use of combinations.
No clinical studies on IL-4 and BV combination therapy.	What are the safety and efficacy outcomes of combining IL-4 blockade and BV in MF patients?	Requires caution	Hinders clinical guideline development; necessitates preclinical and early-phase trial evidence.

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
