# Peer review of "IL-4 and Brentuximab Vedotin in Mycosis Fungoides: A Perspective on Potential Therapeutic Interactions and Future Research Directions"

_cimb, 2025, doi:10.3390/cimb47080586_

Round 1
Reviewer 1 Report
Comments and Suggestions for Authors
This manuscript proposes a novel therapeutic hypothesis: co-targeting the IL-4/Th2 pathway and CD30 in CD30-positive mycosis fungoides (MF). To my knowledge, this specific combination (IL-4 pathway inhibition alongside Brentuximab Vedotin, BV) has not been previously reported in clinical or preclinical studies, underscoring its novelty. The authors make a compelling case that MF’s Th2-skewed microenvironment – characterized by IL-4 and related cytokines – contributes to tumor progression and resistance. Given BV’s established activity in CD30⁺ T-cell lymphomas but known issues with resistance/relapse, the idea of simultaneously modulating the microenvironment (via IL-4 blockade) to enhance BV efficacy is highly significant. If validated, this strategy could address an unmet need by improving response durability and overcoming immune evasion in MF. The authors appropriately frame the hypothesis as biologically compelling but unproven, emphasizing that no direct evidence yet supports it. This candid acknowledgment is important. Overall, the rationale is sound and of clear interest: it targets both the malignant cells and the supportive milieu, aligning with modern combination therapy principles.
Overall, this is a well-crafted and insightful manuscript that meets the standards for publication in Current Issues in Molecular Biology. The topic is timely and novel, the scientific content is strong, and the writing is clear and professional. I do not see any major flaws that would require a fundamental reorganization or rethinking of the work (hence, not a “major revision” or rejection). Instead, I recommend minor revisions focused on polishing and clarifying the presentation.
Justification: The minor issues noted – such as occasional awkward phrasing, very long sentences, and the need to ensure all data points are cited – can be addressed with a careful editing pass. For example, improving specific wording (“fatal effect” → “cytotoxic effect”, “combination (ADC)” → “conjugate (ADC)”) and breaking up dense paragraphs will enhance readability without altering the science. It would also be beneficial for the authors to slightly temper a few speculative statements to maintain a cautious scientific tone. Additionally, verifying that all factual claims (e.g., response rates, case reports) are accompanied by references will strengthen the manuscript’s credibility. These changes are relatively straightforward.
Importantly, the manuscript’s core hypothesis and review of literature are very solid. The authors have demonstrated comprehensive knowledge of MF immunobiology and thoughtfully integrated that with a therapeutic perspective. The suggestions for future research are a highlight and will likely inspire further studies – exactly the purpose of a perspective article. Provided the authors implement the minor edits above and double-check the manuscript for any consistency issues, I am confident that the paper will be publication-ready.
In conclusion, I commend the authors for addressing a complex question in a rigorous manner. I encourage them to revise the manuscript per the points above. After minor revisions to refine the language and presentation, I fully support the acceptance of this article for publication in Current Issues in Molecular Biology.
Author Response
Thank you for taking the time to extensevely review our paper and pointing out some issues! We agree that in the process of writing the article, we may have gotten a bit carried away with the ideas we were chasing and managed to produce less than perfectly readable text. Here are the changes we did to address the issues you pointed out : Comment 1 : "For example, improving specific wording (“fatal effect” → “cytotoxic effect”, “combination (ADC)” → “conjugate (ADC)”) and breaking up dense paragraphs will enhance readability without altering the science." Response 1: 1.1. Fixed/improved specific wording : - "fatal effect " the fatal effect on malignant cells -> cytotoxic effect - "antibody-drug combination" antibody-drug combination (ADC) -> antibody-drug conjugate(ADC) 1.2. Slight tone alterations - "immune reprogramming " (aim : reduce speculative tone) Position: page 11, table What : Does IL-4 pathway inhibition enhance the efficacy of BV in MF through immune reprogramming or modulation of CD30? Changed to : Does IL-4 pathway inhibition enhance the efficacy of BV in MF by modulating immune activity or altering CD30 expression? - "transform outcomes" (aim :tone down) Position: page 8. What : "the potential to transform outcomes for patients with CD30-positive MF" Changed to : "the potential to improve outcomes for patients with CD30-positive MF " - "worthy pursuit" (aim : keep more cautious scientific tone) Where: page 8 What : "makes this a worthy pursuit" Changed to : "warrants further investigation" 1.3. Breaking dense paragraphs into more readable langauige: - Section 3.1.2 – IL-4 in MF-specific pathogenesis (page 4) We split the overly long paragraph starting with "Additional evidence links.." - Section 3.3 — Theoretical Framework for Combination Therapy (page 7) We split the paragraph starting with "Intriguingly.." - Section 3.4.1 — Knowledge Gaps : slichtly edited to make a paragraph break clearer. - Section 3.4.2 — Proposed Research Framework : split the overlong paragraph at the beginning Comment 2: "The minor issues noted – such as occasional awkward phrasing, very long sentences[..]" Response 2: Thank you once again for your observation. We re-read the article after receiving your notes and yes, some of the sentences were overly complex and sometimes unnecessarily burdened with sub clauses.Here are the edits we did to address this issues: Awkward phrasing, very long sentences: - Section 3.3 – Theoretical Framework shortened/split long sentence starting with "This apparent contradiction reflects the complexity.." Changed to : "This contradiction reflects the complex and context-dependent regulation of CD30. It may be influenced by the differentiation state of the cells, concurrent signaling pathways, the duration and intensity of IL-4 exposure, and surrounding cytokine activity." - Section 3.4.1 – Knowledge Gaps shortened simplified sentence starting with "We currently lack a precise understanding.." Changed to : "It remains unclear how IL-4 signaling—or its inhibition—affects CD30 expression in human MF cells. This includes both direct effects and broader microenvironmental influences. These mechanisms likely extend beyond simple receptor-level regulation." - Section 3.1.3 – IL-4/STAT6/CD30 Axis shortened/simplified sentence starting with "This dual regulation means that the net effect ..." Changed to : "The effect of IL-4 on CD30 is context-dependent. Factors such as cell type, exposure timing, and local signaling cues determine whether CD30 is up- or downregulated. This variability directly affects how well CD30-targeted therapies like Brentuximab Vedotin perform, influencing both initial drug response and the emergence of resistance - Section 3.4.2 – Research Framework simplified long sentence starting with "These models will be employed to assess the efficacy, safety (including potential synergistic toxicities.." Changed to : "These models will assess the combination’s efficacy, safety, and pharmacologic profile—especially the risk of neurotoxicity. Importantly, preclinical work must also evaluate the possibility of paradoxical effects, including disease acceleration. This concern has emerged with IL-4 inhibitors in some clinical contexts." - Section 3.4.1 – Knowledge Gaps simplified sentence starting with "Understanding how modulating the IL-4 pathway in conjunction.." Changed to : "It is still unclear how IL-4 pathway modulation—when combined with BV—affects anti-tumor immunity and long-term surveillance. This question is central to predicting long-term disease control and patient outcomes." - Section 3.3 – Theoretical Framework simplified sentence starting with "Primary resistance may result from.." Changed to : "Primary resistance may stem from low baseline CD30 expression or IL-4-driven downregulation that limits BV binding. Over time, chronic IL-4 exposure could select for CD30-low clones or trigger adaptive mechanisms that reduce BV sensitivity" - Section 3.4.1 – Resistance Mechanisms simplified sentence starting with "Potential resistance mechanisms requiring investigation include compensatory .." Changed to : "Potential resistance mechanisms include: compensatory increases in IL-13 or IL-5 after IL-4 blockade; selection for CD30-negative clones; activation of alternative survival pathways; altered MMAE metabolism or drug efflux; and TME remodeling that limits antibody access.”
Reviewer 2 Report
Comments and Suggestions for Authors
The article conducted by Mihaela Andreescu et al is very interersting review regarding perspective on potential therapeutic interactions and future research directions on IL-4 and Brentuximab Vedotin in Mycosis Fungoides. However in my opinion manuscript needs some minor changes in order to improve its quality.
The purpose of the work has not been defined sufficiently. The authors should focus more on the research problem, explain why they took up this topic, why this topic is so important to them.
The methodological chapter is well written. The methods used are well explained.
Conclusion - inference should be clearly presented and consist an inspiration for future research, provide suggestions for future studies. Authors should clearly explain why tehy decided to analyze this problem.
Author Response
Thank you for your review. We appreciate your interest and are pleased that you found the proposed article valuable.
Comments 1:
"The purpose of the work has not been defined sufficiently. The authors should focus more on the research problem, explain why they took up this topic, why this topic is so important to them"
Response 1:
We also appreciate your observation regarding the need to clarify the motivation behind this study.
To address this point, we have added the following explanation to the end of the Introduction:
"As clinicians, we routinely encounter patients who illustrate what has already been established: MF is both frequent and difficult to treat effectively in the long term. In advanced or treatment-refractory cases, median survival can drop to 2–4 years, with quality of life significantly impaired by skin failure, infections, or systemic symptoms. The need to expand available therapeutic options is not theoretical—it is urgent and ongoing. This review was motivated by the observed intersection between IL-4–driven immune dysregulation and the partial success of CD30-targeted therapy in MF. Despite the potential for interaction, these two axes have never been explored in tandem. Here, we aim to provide a mechanistic rationale to support future investigation of this untested therapeutic combination."
We believe this addition now clearly communicates the rationale for undertaking this review and why this topic is of clinical and scientific importance.
Comments 2:
"Conclusion - inference should be clearly presented and consist an inspiration for future research, provide suggestions for future studies. "
Response 2:
Thank you again. We understand the importance of providing a sharper inference in the Conclusion to give readers a clear takeaway.
We also agree that explicitly restating key research directions would strengthen the closing section.
To address this, we revised the final paragraph of the Conclusion as follows:
"Despite these complexities, the hypothesis remains biologically plausible and clinically relevant. Future research should prioritize preclinical models to test the safety and efficacy of combining IL-4 pathway inhibition with BV, with particular focus on tumor mi-croenvironment remodeling, CD30 expression kinetics, and cytotoxic immune function. Biomarker development—including STAT6 activation, GATA3 expression, and CD30 dynamics—may further guide patient selection and treatment sequencing in earlyphase trials."
We hope this improved structure and focus provide a more inspiring and actionable conclusion, as suggested.